# Development and validation of scales for speaking self-efficacy: Constructs, sources, and relations

**Yawen Wang, Peijian Paul Sun**[ORCID]*

Department of Linguistics, Zhejiang University, Hangzhou, China

* luapnus@zju.edu.cn

## Abstract

Speaking is not a compulsory language skill assessed in the English subject of the National College Entrance Examination in China. This explains why, in elementary and secondary schools, less focus has been placed on the development of English-speaking abilities among Chinese students, resulting in their unbalanced mastery of language skills. Although self-efficacy is a crucial factor influencing students' language performance, our understanding of speaking self-efficacy is insufficient in terms of its construct, its sources, and the relationships between the two. We, therefore, constructed psychometrically sound instruments to measure speaking self-efficacy, including the EFL Speaking Self-Efficacy Scale (EFL-SSES) and the EFL Sources of Speaking Self-Efficacy Scale (EFL-SSSES), in accordance with Bandura's 1986 self-efficacy theory. Additionally, we performed path analysis to figure out the relationship between the construct and the sources of speaking self-efficacy. The results revealed the key role of physiological and emotional states and marginal importance of vicarious experience for speaking self-efficacy, advancing our grasp of self-efficacy theory in the speaking domain. Our research sheds valuable light on how to assist researchers and educators in identifying and enhancing students' speaking self-efficacy via a variety of sources.

## 1 Introduction

In China, English as a foreign language (EFL) learners' listening and speaking skills are often weaker than their writing and reading skills [1]. This phenomenon is mainly ascribed to the following two reasons. First, the monolingual context without an authentic English language environment cannot provide learners with the opportunities necessary for developing their English-speaking competence. Second, pedagogical methods in middle and high schools in China have placed a low premium on speaking, resulting in learners' poor English-speaking abilities [2]. Consequently, students may experience anxiety and vulnerability while expressing their thoughts, engaging in debates, or interacting with others in English. To mitigate this issue, one possible solution is to enhance students' confidence in EFL speaking, given that students with higher self-efficacy tend to set more ambitious goals, exert greater effort in completing academic assignments, and employ a variety of flexible and diverse learning strategies [3,

**Funding:** The work was supported by the National Social Science Foundation of China awarded to Dr. Peijian Sun (Grant no. 19CYY008). The funder had no role in study design, data collection and analysis, decision to publish, or preparation of the manuscript.

**Competing interests:** The authors have declared that no competing interests exist.

4]. In this study, the English speaking construct refers to English learners' ability to express themselves in both monologic and interactive ways and to collaborate in the development and maintenance of interaction [5].

Since Bandura [6] first proposed the concept of self-efficacy, researchers have been actively investigating the influence of learners' self-efficacy on their learning process or outcomes [7, 8]. However, self-efficacy is situation dependent. Therefore, concentrating on a specific context and activity domain would advance our understanding of self-efficacy [6]. Although numerous measures of EFL learners' self-efficacy have been developed [9], research related to self-efficacy in the domain of speaking remains scarce. Furthermore, Bandura [10] proposed that people acquire self-efficacy by interpreting information from four distinct sources. Nevertheless, there is a dearth of research on sources of speaking self-efficacy among EFL learners. More attempts, therefore, are in urgent need to elucidate the construct and sources of speaking self-efficacy and build robust instruments to deepen our knowledge of speaking self-efficacy. To bridge the gaps, we developed and validated scales for speaking self-efficacy in an attempt to enhance our understanding of this construct and its sources, as well as to provide pedagogical insights for teachers to foster students' self-efficacy in EFL speaking. Specifically, we attempt to address the following three research questions (RQs).

RQ1: What is the construct of speaking self-efficacy?

RQ2: What is the status quo of EFL learners' speaking self-efficacy and its sources?

RQ3: What is the relationship between the construct and the sources of speaking self-efficacy?

## 2 Literature review

### 2.1 Theoretical foundation of self-efficacy: Social-cognitive theory

Bandura's social cognitive theory posits that learners are agents who "draw on their knowledge and cognitive and behavioral skills to monitor their actions [and] enlist cognitive guides and self-incentives to produce desired results" [10]. Derived from the social cognition theory, self-efficacy refers to a person's judgment of his or her capabilities to complete a specific task with his or her skills [6, 11]. As a personal factor in the form of beliefs, it has a major influence on how individuals approach goals, initiatives, and obstacles [12, 13]. Students who have a higher level of self-efficacy tend to show positive orientations toward academic tasks. They are eager to solve new issues and willing to persevere in the face of adversity. Consequently, their self-efficacy can be largely reinforced and improved. In contrast, people with poor self-efficacy constantly doubt their abilities to perform tasks and often avoid dealing with difficulties and challenges.

According to social cognitive theory, human beings have cognitive abilities to self-organize, self-reflect, and self-regulate in response to the changes in the environment and determine their own social destiny. The core of the social cognitive theory lies in "triadic reciprocality", in which internal personal factors, behavioral patterns, and environmental events influence each other bidirectionally [10]. Individuals must take a proactive role in their growth and make things happen through their own efforts. In other words, the interaction of personal, environmental, and behavioral events determines how individuals behave [14].

Bandura [12] argues that the operational function of self-efficacy depends on an essential belief linking human agency and efficacy beliefs. Individuals' perceived self-efficacy may be contextual [15, 16], implying that the characteristics of specific tasks or situations even in one particular domain may influence their self-efficacy. Domain-specificity is a critical feature of self-efficacy [17], indicating that individuals' self-efficacy levels vary across different domains

of activity. It is determined based on individuals' perceived competence to accomplish specific tasks in particular situations [18]. For instance, an EFL learner may be quite confident in his or her writing abilities, but may have poor self-efficacy in speaking. Thus, it is indispensable to explore self-efficacy by concentrating on a specific context and activity domain and developing questionnaires accordingly.

Studies on students' self-efficacy have received continual interest in the field of language acquisition [19, 20]. Self-efficacy is a significant predictor of academic achievement in general [7, 8, 21] and language proficiency in particular [22, 23]. Recently, Wang and Sun [24] resorted to meta-analysis to confirm a positive relationship between self-efficacy beliefs and language learning outcomes. This comprehensive review suggested that research regarding self-efficacy is thriving across diverse domains (overall, receptive and productive language skills), demonstrating a domain-specific feature of self-efficacy. This meta-analysis indicated that few studies examined self-efficacy in listening and speaking compared to other domains, which may be accounted for by the fact that schools place a premium on students' literacy development, whilst neglect listening and speaking [25]. Therefore, more investigations into learners' self-efficacy in second language speaking is needed to enrich our understanding of its internal mechanism.

## 2.2 Constructs and measures of speaking self-efficacy

According to Bandura [6], researchers would find the most utility from self-efficacy by focusing on a specific context and activity domain. Given that self-efficacy is a domain-specific and multi-dimensional construct [15], it is necessary to examine self-efficacy by placing it within situation-specific contexts.

In the domain of EFL learning, self-efficacy has found to be predictive of listening [26], reading [27], and writing [28–30]. However, previous research often adopted general self-efficacy construct to represent learners' language-skill-specific self-efficacy, resulting in biased findings [18, 31]. Speaking, for example, as a highly complex skill, requires (1) knowledge of language and discourse (pronunciation, grammar, vocabulary, and discourse); (2) fundamental speaking skills (chunking, signaling intent, and turn-taking); and (3) communication strategies (paraphrasing, rephrasing, and approximation). Therefore, it will be inappropriate to use general self-efficacy items to measure speaking self-efficacy. Rather, speaking self-efficacy should be examined from a multi-dimensional perspective, given the complex nature of speaking skill. In this study, speaking self-efficacy is defined as the confidence in one's speaking ability to properly perform grammar, usage, communication, and interaction.

In terms of self-efficacy construct and/or measures, there is an increasing number of studies that have been conducted to establish valid and reliable instruments for measuring self-efficacy. For instance, the Motivated Strategies for Learning Questionnaire (MSLQ) is the most frequently used instrument [32]. The 8-item "Self-Efficacy for Learning and Performance" is one of the 15 subscales. Eight questions were developed to ascertain students' perceptions of their ability to succeed in an undergraduate course. Since its release, MSLQ has been widely examined for its psychometric qualities [e.g., 32, 33]. Apart from MSLQ that conceptualizes self-efficacy from a broad perspective, Wang [9] developed the Questionnaire of English Self-Efficacy (QESE) with 32 items to measure EFL learners' self-efficacy beliefs. Based on Bandura's [10] guidelines, the QESE met the demand for validity and reliability to probe EFL learners' self-efficacy beliefs. Each item demands students to judge their abilities to complete specific tasks in English listening, speaking, reading, and writing. The QESE is the most extensively used self-efficacy scale in the EFL field [19, 20, 34].

Among the four language skills, writing receives the most attention from researchers, resulting in most measures available for assessing writing self-efficacy. For example, Teng et al., [35] developed the Second Language Writer Self-Efficacy Scale for college students based on self-regulated learning theory and social cognitive theory. This 7-point Likert scale contains 20 items and has demonstrated satisfactory psychometric properties in terms of internal and composite reliability, convergent validity, and discriminant validity. Bruning et al., [22] also constructed the Self-Efficacy for Writing Scale to measure middle and high school students' writing self-efficacy. This 0–100 scale consists of 16 items in three writing-related dimensions. F factor analysis (CFA) confirmed the proposed 3-factor structure.

However, to the best of our knowledge, only a very limited number of studies have attempted to develop questionnaires to measure students' speaking self-efficacy [36–40]. For example, Sundari and Dasmo [37] adapted from Bandura's [17] guide to construct a 10-item speaking self-efficacy scale, ranging from 0 to 100. However, this research had a limited sample size and lacked validity and reliability evidence for the self-efficacy construct measures used. Additionally, Asakereh and Dehghannezhad [39] proposed the Speaking Skills Self-Efficacy Beliefs questionnaire, a 28-item 5-point Likert scale. Nevertheless, this scale only provided reliability results and lacked evidence of structural validity.

Recently, Harris [36] developed the Communicative Self-Efficacy Questionnaire. This 6-point Likert scale is comprised of 16 items for measuring tertiary students' foreign language self-efficacy concerning their speaking and listening skills. This scale was analyzed through the Rasch model. This study focused more on learners' communicative skills rather than their speaking skills. Li and Sui [40] constructed and validated the 18-item English Speaking Self-Efficacy Scale for Chinese College Students. This 11-point Likert scale was based on Zimmerman et al.'s construct of self-efficacy [41] and the multidimensional framework of complexity, accuracy and fluency. It primarily assesses students' speaking self-efficacy in terms of complexity, accuracy, and fluency. The abovementioned research limitations suggest that the construct of speaking self-efficacy is promising for further investigation.

As mentioned earlier on, self-efficacy is task-specific and should be analyzed accordingly [17]. However, due to the mix use of general and task-specific self-efficacy in EFL research, the findings of self-efficacy display inconsistency. Additionally, there is a paucity of instruments to measure speaking self-efficacy in EFL settings regarding language skills. Given that an EFL learner's self-efficacy to build reading abilities is different from that required to develop speaking skills, it is crucial to adopt a narrower perspective to explore the distinct development process of each language skill. Considering there is a paucity of instruments to measure speaking self-efficacy in EFL settings, this study aims to fill this gap.

## 2.3 Sources of speaking self-efficacy

According to social cognitive theory, expectations of personal efficacy are based on four sources of self-efficacy: mastery experience (ME), vicarious experience (VE), social persuasion (SP), physiological and emotional states (PES) [6]. ME as the most influential factor of self-efficacy beliefs is the interpreted result of one's own previous accomplishments [11]. Success in overcoming difficult EFL tasks boosts a person's self-efficacy, while repeated failures erode it. Interpretation of the outcomes of their efforts may have long-lasting effects on their self-efficacy [10, 42].

Observing others' performances, known as VE, has a remarkable impact on self-efficacy [6]. Individuals evaluate their own chances of success or failure based on their observations of others who excel at or struggle with the same or comparable EFL tasks. Successful actions of others often boost one's self-efficacy, and vice versa. It is worth mentioning that the classroom

climate also greatly impacts students' vicarious experience [43]. Specifically, students' self-efficacy decreases when they engage in vicarious learning in competitive classrooms, while there is no significant change in students' self-efficacy in non-competitive classrooms.

The third source of self-efficacy is SP [6]. Encouragement and positive feedback would strengthen EFL learners' self-efficacy, whilst discouragement and negative feedback would deteriorate it. Rather than directly generating self-efficacy beliefs, social persuasion assists in the development of self-efficacy when combined with other strategies [10, 11]. Its impact is intimately attributed to the credibility and reputation of the information source. As previous research suggests, it may be easier to undermine an individual's self-efficacy via social persuasions than to enhance it [44]. Furthermore, social persuasion has a more substantial effect on learners' self-efficacy in collectivist cultures than in individualist cultures [12, 45].

Last but not least, PES have a significant influence on the development of self-efficacy [6]. How individuals perceive their emotional reactions is of prime importance to their perceptions of themselves [10]. If a speaker lives in dread in anticipation of his or her speaking performance, he or she is likely to take it as evidence of inadequate speaking ability. Increasing individuals' physiological and emotional well-being and reducing negative emotional states may strengthen self-efficacy.

Bandura's [6, 10] theoretical model of four sources of self-efficacy has been employed in a plethora of research to demonstrate its latent structure and predictive roles in different academic domains [11]. However, few instruments have been developed particularly to assess EFL learners' sources of speaking self-efficacy. Inadequate research has been conducted to ascertain the relationships between speaking self-efficacy and its sources among EFL learners. Only Zhang et al. [46] investigated the self-efficacy of English public speaking and its sources. They developed the 12-item English Public Speaking Self-Efficacy Scale and 14-item Sources of EPS Self-Efficacy Scale on 5-point Likert scales. It was found that there barely existed any direct or indirect effect on English public speaking performance via English language proficiency and self-efficacy sources, though the four theoretical sources had different effects on self-efficacy [38]. Given that the four sources serve as critical antecedents of self-efficacy, there exists the demand to further examine their relationships in the domain of speaking. Our present study serves as an initial step to develop and preliminarily validate scales for speaking self-efficacy with an attempt to understand its construct, its sources, and the relationship between them.

## 3 Materials and methods

Informed by the social-cognitive theory [10], we sought to conceptualize and validate the multidimensional structures of speaking self-efficacy and its sources. In Study 1, we developed and validated a quantitative instrument exploring college EFL learners' speaking self-efficacy, namely EFL Speaking Self-Efficacy Scale (EFL-SSES). It went through three phases: (a) item generating, (b) exploratory factor analysis (EFA), and (c) CFA. In Study 2, we developed and validated another instrument evaluating students' sources of speaking self-efficacy, namely EFL Sources of Speaking Self-Efficacy Scale (EFL-SSSES), based on the four sources proposed by Bandura [6]. It also went through the same three phases. In Study 3, we examined the relationships between speaking self-efficacy and its sources with path analysis. Please note that written informed consent was obtained from all the participants before any data collection.

### 3.1 Study 1: EFL-SSES development and validation

**3.1.1 Phase 1: Item generation.** The process of item generation began with individual interviews. Ten undergraduate students were invited to voluntarily participate in the semi-

structured interview with different gender, year level, and disciplinary major. The interview protocol was designed based on previous literature [9, 32]. S1 Appendix presents the analysis procedure of the interview data, which helped to generate the initial items. The initial item pool included 30 speaking self-efficacy items. Since the original items were in Chinese, a method of forward-backward translations was used by both authors to ensure the conceptual equivalency of the English and Chinese language versions.

Two professors in the field of second language acquisition (SLA) were invited to scrutinize the content validity of the initial item pool. These confusing and redundant items were reworded or eliminated based on their comments. The examination resulted in the deletion of 8 items, leaving 22 items for final analysis. For instance, one item read "*I can produce sentences with idiomatic expressions.*" Since this behavior was uncommon for speakers, it was therefore deleted in the evaluation process.

The first version of the questionnaire was pilot-tested with 40 participants who exhibited a diversity in terms of gender, year level, and disciplinary major and had taken courses related to oral English. This led to the reformulation of some items for clarity and readability, eliminating three irrelevant and double-barreled items and rewording two others. This resulted in the formation of a preliminary 19-item scale with a 7-point Likert scale ranging from "1 = strongly disagree" to "7 = strongly agree".

**3.1.2 Phase 2: EFA of EFL-SSES.** *Participants*. A total number of 242 undergraduate students from one southeast university in China voluntarily participated in this phase. We recruited participants after midterm based on the convenience sampling and the voluntary participation principle. They were non-English major students with Mandarin Chinese as their native language. Questionnaires were administered during their college English course. Participants were briefed on the purpose of the study and were requested to respond honestly and sincerely to the items. They spent on average 5 minutes completing the randomized Chinese version of the EFL-SSES. Fifteen questionnaires were discarded as the participants did not complete them, leaving 227 valid questionnaires. The response rate was 93.80%. Their average age was 19 years old ($M = 18.94$, $SD = 0.85$), with 43% female ($N = 97$) and 57% male ($N = 130$). They reported an average 11 years of formal English language learning ($M = 10.75$, $SD = 1.76$). Participants came from four majors: Engineering (38%, $N = 87$), Liberal arts (29%, $N = 65$), Science (20%, $N = 45$), and Medicine (13%, $N = 30$).

*Results*. EFA was conducted to test the hypothesized factor structure and refine the items through SPSS version 24 [47]. The mean values of all the items ranged from 3.21 to 4.67. The standard deviations ranged from 0.08 to 0.10, with the skew and kurtosis indices from -0.84 to 0.57 and -0.51 to 0.37 respectively. The data in the present study were considered to be univariate normal. The results of the Kaiser–Meyer–Olkin test verified the sample adequacy (KMO = 0.91). The result of Barlett's test of sphericity ($\chi^2$ (105) = 2595.97, $p < 0.001$) indicated that the correlations between items were appropriate for factor analysis.

To determine the underlying structure of EFL-SSES, the 19 items were subjected to EFA with maximum likelihood analysis and a promax rotation method. Four factors with eigenvalues above 1.0 were extracted. The EFA was repeated until all items in each factor had reasonably high loadings (greater than 0.4) and no more items could be excluded from the analysis. According to the pattern matrix from EFA and the internal consistency reliability results, 15 of the 19 original items in the EFL-SSES loaded high on the four factors, as confirmed by the scree plot with parallel analysis (see Fig 1).

These factors were labelled linguistic self-efficacy (LSE), self-regulatory efficacy (SRE), delivery self-efficacy (DSE), and performance self-efficacy (PSE) based on their contents. The four factors accounted for 75.69% of the variance. The Cronbach's alpha (α) was used to check the internal consistency reliability of each extracted factor. Co-efficiencies of all four factors

**Scree plot with parallel analysis of EFL−SSES**

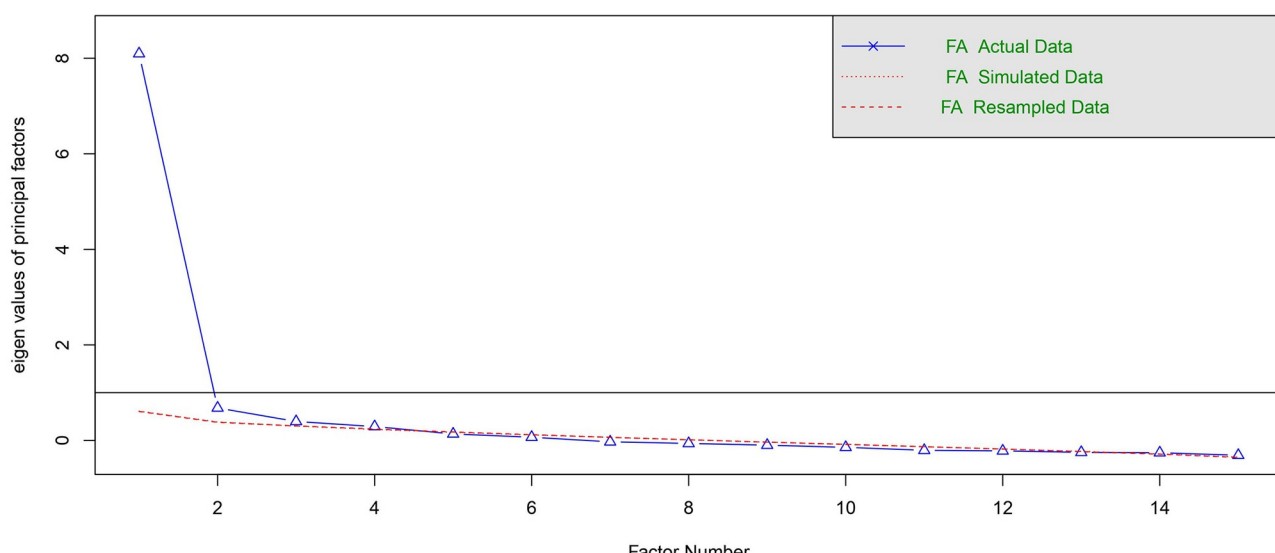

**Fig 1. The scree plot with parallel analysis of EFL-SSES.** *Note.* The point where the simulated eigenvalue (the red lines) meets the corresponding, real eigenvalue (the blue line) suggests the number of factors.

exceeded the threshold of 0.7 (see Table 1). S2 Appendix presents the finalized items of EFL-SSES. LSE refers to students' judgements of their capability to execute various phonological, lexical, syntax, and discourse skills necessary to communicate effectively in English in the classroom. SRE is associated with EFL learners' perceived capability to execute metacognitive strategies while speaking English in class. DSE refers to EFL learners' speaking ability to talk with emotional appeal by using self-control in the classroom. PSE represents students' perceptions of their speaking capability to accomplish course assignments or comprehension of course material.

**3.1.3 Phase 3: CFA of EFL-SSES.** After administering the EFA analysis, CFA was conducted to determine the factorial validity of the 15-item EFL-SSES via SPSS [47] and AMOS 24 software [48]. The model fit was evaluated based on multiple fit indices, including $\chi^2$ significance test, Comparative Fit Index (CFI), the Goodness of Fit Index (GFI), the Tucker–Lewis Index (TLI), the Standardized Root Mean Square Residual (SRMR), and the Root Mean Square Error of Approximation (RMSEA). For a model with a sample size larger than 250 and its observed variable number greater than 30, CFI, TLI, and GFI above 0.92, SRMR below 0.08, and RMSEA below 0.07 indicate adequate fit [49]. The reliability of the scale was examined by Cronbach's alpha ($\alpha$). Average variance extracted (AVE) and composite reliability (CR) were used to test the convergent validity of the scale. The convergent validity of the construct is acceptable if AVE is larger than 0.5 and CR is higher than 0.6 [50]. The heterotrait-monotrait

**Table 1. Results of EFA and reliabilities of the four-factor EFL-SSES.**

| Factor | Items | Factor loadings | Total Variance Explained | Cronbach's alpha ($\alpha$) |
|---|---|---|---|---|
| LSE | 5 | 0.549–0.895 | 7.78% | 0.884 |
| SRE | 3 | 0.414–0.871 | 5.45% | 0.780 |
| DSE | 3 | 0.564–0.980 | 5.56% | 0.907 |
| PSE | 4 | 0.627–0.932 | 56.90% | 0.893 |

ratio of correlations (HTMT) proposed by Henseler et al. [51] was used to test the divergent validity. If the HTMT score is less than 0.90, discriminant validity between two reflective notions has been established.

*Participants.* To conduct CFA, the scale was administered to 300 non-English major undergraduate students with Mandarin Chinese as their native language. They were recruited based on convenience sampling and did not take part in the previous EFA phase. Questionnaires were administered during their college English course. Participants spent on average 5 minutes completing the randomized Chinese version of the EFL-SSES. The questionnaires from 11 students were disregarded because of missing values and inconsistent responses to care-check items. This reduced the number of participants to 289. The response rate was 96.33%. They consisted of 170 males (59%) and 119 females (41%). Their age ranged from 17 to 21 with an average of 18.80 (*SD* = 0.68). They reported an average 11 years of formal English language learning (*M* = 10.79, *SD* = 1.88). All participants were from four majors: Engineering (37%, *N* = 107), Science (30%, *N* = 86), Liberal arts (23%, *N* = 66), and Medicine (10%, *N* = 30).

*Results.* CFA was conducted to validate the underlying factor structure. The mean score of all 15 items ranged from 3.50 to 5.18, with standard deviations ranging from 1.12 to 1.41. The data were univariate normal, with skewness ranging from -0.84 to 0.30 and kurtosis values ranging from -0.52 to 0.90. Based on the theoretical understanding of self-efficacy, a zero-order correlated model was proposed (Model 1). Maximum likelihood estimation was used to estimate the model. This model assumed that the underlying four factors of speaking self-efficacy were correlated with each other The CFA analysis demonstrated a reasonable fit to the data, $\chi^2(84) = 199.785$, $p < .001$, CFI = 0.963, GFI = 0.918, TLI = 0.954, SRMR = 0.0405, and RMSEA = 0.069 [0.057~0.082]. Fig 2 showed the standardized regression weights of Model 1.

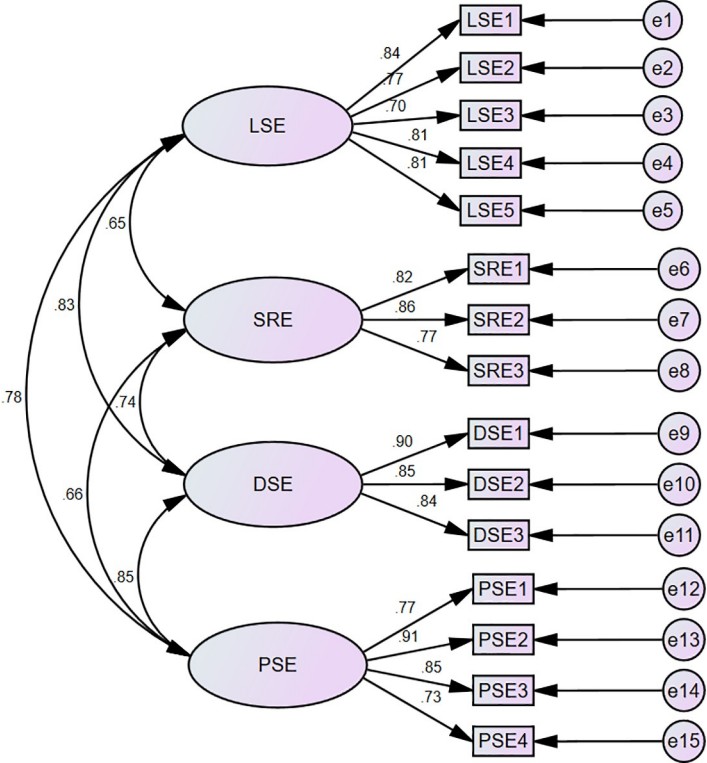

**Fig 2. Four-factor correlated model of speaking self-efficacy.** *Note. N* = 289. LSE = linguistic self-efficacy; SRE = self-regulatory efficacy; DSE = delivery self-efficacy; PSE = performance self-efficacy. All item parameter estimates and latent variables correlations were significant (*p* < .001).

**Table 2. Results of convergent and discriminant validity of the four-factor EFL-SSES.**

| Factor | HTMT | | | | Cronbach's alpha (α) | CR | AVE |
|---|---|---|---|---|---|---|---|
| | 1 | 2 | 3 | 4 | | | |
| LSE | | | | | 0.89 | 0.83 | 0.51 |
| SRE | 0.65 | | | | 0.85 | 0.86 | 0.67 |
| DSE | 0.83 | 0.75 | | | 0.90 | 0.78 | 0.56 |
| PSE | 0.80 | 0.68 | 0.86 | | 0.88 | 0.83 | 0.57 |

**Table 3. Goodness-of-fit indices for competing models of EFL-SSES.**

| Model | ML $\chi^2$ | df | $\chi^2$/df | CFI | GFI | TLI | SRMR | RMSEA | RMSEA 90% CI |
|---|---|---|---|---|---|---|---|---|---|
| Model 1 | 199.785 | 84 | 2.378 | 0.963 | 0.918 | 0.954 | 0.0405 | 0.069 | 0.057–0.082 |
| Model 2 | 617.174 | 90 | 6.857 | 0.833 | 0.75 | 0.805 | 0.0695 | 0.143 | 0.132–0.153 |
| Model 3 | 879.375 | 90 | 9.771 | 0.749 | 0.697 | 0.708 | 0.4296 | 0.175 | 0.164–0.185 |

The items were all significantly related to their underlying constructs and the factor loadings ranged from 0.513 to 0.815. Cronbach's alpha coefficient indicated a robust internal reliability of the scale. The results in Table 2 prove the convergent and discriminant validity of EFL-SSES.

To further verify Model 1, two alternative conceptualizations of speaking self-efficacy models were proposed for comparison. Model 2 hypothesized a unidimensional model of speaking self-efficacy in which all the items loaded on one factor. Model 3 hypothesized a four-component uncorrelated model of speaking self-efficacy. That is, the four components of speaking self-efficacy, LSE, SRE, DSE, and PSE, are not correlated with each other. The model fit indices of the three models were compared (Table 3). The comparison results revealed that Model 1 had the best fit indices when compared to Model 2 and Model 3, providing forceful support to the construct validity of the four-dimension correlated structure of speaking self-efficacy (Model 1).

### 3.2 Study 2: EFL-SSSES development and validation

**3.2.1 Phase 1: Item generation.** The process of item generation was guided by Bandura's theoretical work, which proposes the four sources of self-efficacy [6]. After examining all the available literature on sources of self-efficacy [13, 46], the initial 30 items were constructed. The questionnaire items were reworded to accommodate EFL learners' speaking learning process. Given the level of English language proficiency of potential participants, the English scale was translated into Chinese by the first author. In order to prevent potential loss of meaning, the first and second authors as native Mandarin Chinese and proficient English bilingual speakers carefully examined the items and their Chinese translations back and forth.

Two university professors with expertise in SLA and extensive experience teaching EFL speaking were invited to review the initial pool of items for relevance, clarity, redundancy and readability. Based on their suggestions, 10 items that were deemed confusing, irrelevant, and difficult to comprehend were eliminated, leaving 20 items for the final analysis. It was piloted with 35 students, diverse in terms of gender, year level, and disciplinary major, who had attended oral English courses. Following their feedback, four items were deleted. The final questionnaire consists of 16 items with a 7-point Likert scale.

**3.2.2 Phase 2: EFA of EFL-SSSES.** *Participants.* Convenience sampling was employed, and participants consisted of 230 non-English major undergraduate students with Mandarin Chinese as their native language. Questionnaires were administered during their college English course. Participants spent on average 5 minutes completing the randomized Chinese version of the SSSER. Six students' responses were disregarded because of missing values and inconsistent answers to care-check items. This reduced the number of participants to 224, with 138 males (62%) and 86 females (38%). The response rate was 97.39%. They ranged in age from 16 to 21 years with a mean of 18.8 ($SD$ = 0.72). They reported an average of 11 years' experience studying English ($M$ = 10.74, $SD$ = 1.90). Participants came from four majors: Science (33%, $N$ = 74), Engineering (32%, $N$ = 72), Liberal arts (21%, $N$ = 48), and Medicine (13%, $N$ = 30).

*Results.* EFA was carried out as Study 1. All items have the mean values from 3.53 to 5.21. The standard deviations varied from 1.22 to 1.50. The skewness and kurtosis indices varied from -0.90 to 0.33 and -0.55 to 1.00 respectively. The data were univariate normal. The KMO test confirmed the sample adequacy (KMO = 0.84). The result of Barlett's test of sphericity ($\chi^2$ (78) = 2703.87, $p$ < .001) indicated that correlations between items were suitable for factor analysis. EFA with maximum likelihood estimation and a promax rotation method was performed. We sought to develop the scale with concise sets of indicators and thereby only retained items with high loadings of 0.40 or above from the initial pool of EFL-SSSES. The EFA based on the 16 items yielded a four-factor solution, as confirmed by Fig 3.

Remaining items loaded adequately on their respective factors. Following that, we eliminated three redundant items with lower loadings on the grounds of parsimony. This step resulted in a more succinct solution, since the remaining 13 items adequately covered Bandura's [6] sources of self-efficacy (see details in Table 4). These items account for 85.77% of the variance in total. Cronbach's alpha coefficient of internal consistency of the subcategories was much higher than 0.70. S3 Appendix presents the finalized items of EFL-SSSES.

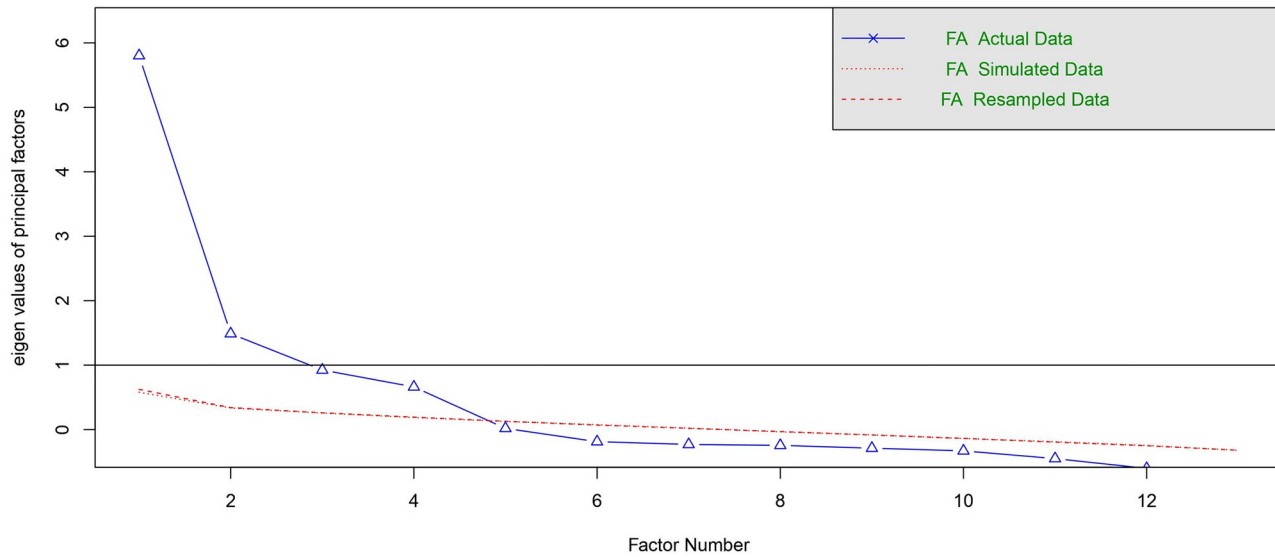

**Fig 3. The scree plot with parallel analysis of EFL-SSSES.** *Note.* The point where the simulated eigenvalue (the red lines) meets the corresponding, real eigenvalue (the blue line) suggests that the number of factors.

**Table 4. Results of EFA and reliabilities of the four-factor EFL-SSSES.**

| Factor | Items | Factor loadings | Total Variance Explained | Cronbach's alpha (α) |
|---|---|---|---|---|
| ME | 4 | 0.695–1.020 | 48.31% | 0.926 |
| VE | 3 | 0.804–1.033 | 17.80% | 0.930 |
| SP | 3 | 0.806–0.894 | 8.56% | 0.911 |
| PES | 3 | 0.817–0.999 | 11.10% | 0.920 |

**3.2.3 Phase 3: CFA of EFL-SSSES.** *Participants.* CFA was implemented through administering the scale generated in the EFA to 307 non-English major undergraduate students whose native language is Mandarin Chinese. They were selected through convenience sampling and based on their willingness to participate in the study, and did not participate in the previous EFA study. Questionnaires were administered during their college English course. Participants spent on average 5 minutes completing the Chinese version of the EFL-SSSES. Questionnaires from 12 students with missing values and inconsistent responses were omitted, which reduced the number of participants to 295 (169 males and 126 females). The response rate was 96.09%. Their age ranged from 17 to 23 with a mean of 18.91 ($SD$ = 0.88). They reported an average 11 years of formal English language learning ($M$ = 10.69, $SD$ = 1.90). All participants were from four majors: Science (38%, $N$ = 111), Liberal arts (27%, $N$ = 80), Engineering (22%, $N$ = 66), and Medicine (13%, $N$ = 38).

*Results.* CFA was implemented to validate the underlying factors using maximum likelihood estimation. The four latent factors were assumed to be correlated and allowed to covary in the model. The mean of all items was between 3.72 and 5.87. Standard deviations varied between 1.04 and 1.50, whereas skew and kurtosis indices ranged from -1.16 to 0.15 and -0.81 to 2.20, respectively. The purpose of this study was to test the four-factor correlated model of speaking self-efficacy sources (Model 4). Each of the indices revealed that the data fitted the model rather well ($\chi^2$(59) = 182.828, $p$ < .001, CFI = 0.957, GFI = 0.912, TLI = 0.943, SRMR = 0.0554, and RMSEA = 0.084 [0.071~0.099]). The standardized regression weights of the four-factor correlated model of sources of speaking self-efficacy are shown in Fig 4. Each item was significantly related to its underlying construct, with factor loadings ranging from 0.628 to 0.925. Cronbach's alpha coefficients indicated that the scale has a robust internal reliability. Table 5 demonstrates the convergent and discriminant validity of the EFL-SSSES.

This study compared the four-factor correlated model of sources of speaking self-efficacy (Model 4) to a null model (Model 5) and an uncorrelated four-factor model (Model 6). In Model 5, we hypothesized that all the 13 items loaded on one overarching factor. This model is supported by the fact that participants perceived all these sources of speaking self-efficacy as a unidimensional construct. Model 6 included four uncorrelated factors representing sources of speaking self-efficacy. A support for this model suggests that mastery experience, vicarious experience, social persuasion, and physiological and emotional states are unrelated constructs. The fit indices for three models were provided in Table 6. The comparison findings revealed that Model 4 had the best fit indices when compared with Model 5 and Model 6, ensuring the construct validity of the four-dimension correlated structure of sources of speaking self-efficacy (Model 4).

## 3.3 Study 3: Relationships between EFL-SSES and EFL-SSSES

Responses from students that participated in both Study 1 and Study 2 went through path analysis to verify the relationship between speaking self-efficacy and its sources.

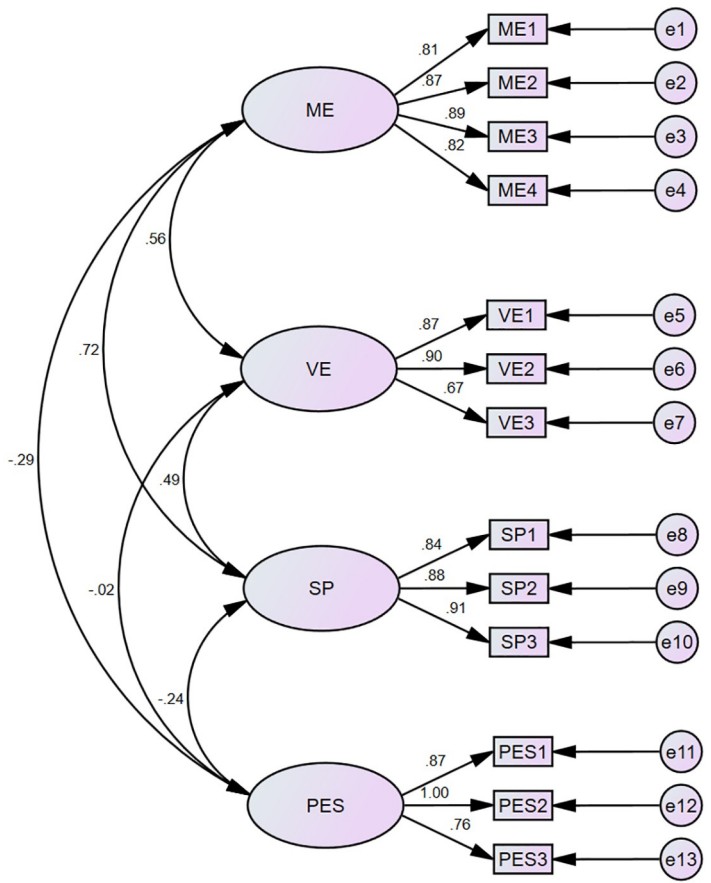

**Fig 4. Four-factor correlated model of sources of speaking self-efficacy.** *Note.* $N$ = 295. ME = mastery experience; VE = vicarious experience; SP = social persuasion; PES = physiological and emotional states. All item parameter estimates and latent variables correlations were significant ($p$ < .001).

**Table 5. Results of convergent and discriminant validity of the four-factor EFL-SSSES.**

| Factor | HTMT | | | | Cronbach's alpha (α) | CR | AVE |
|---|---|---|---|---|---|---|---|
| | 1 | 2 | 3 | 4 | | | |
| ME | | | | | 0.91 | 0.89 | 0.67 |
| VE | 0.57 | | | | 0.84 | 0.85 | 0.66 |
| SP | 0.72 | 0.50 | | | 0.91 | 0.91 | 0.76 |
| PES | -0.34 | 0.02 | -0.27 | | 0.90 | 0.91 | 0.78 |

**Table 6. Goodness-of-fit indices for competing models of EFL-SSSES.**

| Model | ML $\chi^2$ | df | $\chi^2$/df | CFI | GFI | TLI | SRMR | RMSEA | RMSEA 90% CI |
|---|---|---|---|---|---|---|---|---|---|
| Model 4 | 182.828 | 59 | 3.099 | 0.957 | 0.912 | 0.943 | 0.0554 | 0.084 | 0.071–0.099 |
| Model 5 | 1371.852 | 65 | 21.105 | 0.548 | 0.578 | 0.457 | 0.1642 | 0.262 | 0.250–0.274 |
| Model 6 | 472.090 | 65 | 7.263 | 0.859 | 0.798 | 0.831 | 0.2934 | 0.146 | 0.134–0.158 |

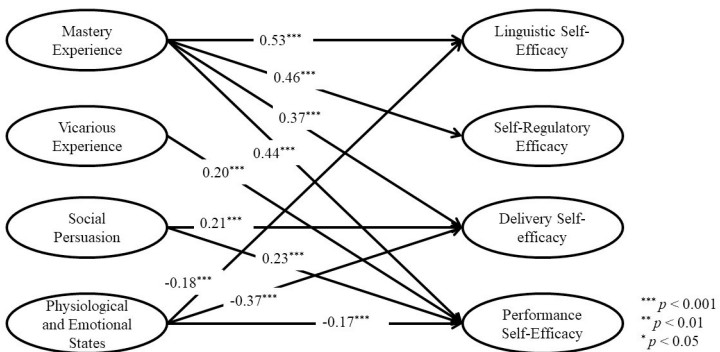

**Fig 5. Path analysis and path coefficients of speaking self-efficacy and its sources.**

**3.3.1 Participants.** There were 304 students that participated in both Study 1 and Study 2. Their average age was 19 years old ($M = 18.87$, $SD = 0.79$), with 61% male ($N = 184$) and 39% female ($N = 120$). They reported having studied English for an average of 11 years ($M = 10.75$, $SD = 1.90$). They came from four majors: Engineering (34%, $N = 103$), Science (27%, $N = 82$), Liberal arts (26%, $N = 80$), and Medicine (13%, $N = 39$).

**3.3.2 Results.** Path analysis was conducted to ascertain the predictive power of each underlying source in the four aspects of speaking self-efficacy. The goodness of fit indices was indicative of an acceptable model-to-data fit ($\chi^2(333) = 896.755$, $p < .001$, CFI = 0.920, GFI = 0.821, TLI = 0.909, SRMR = 0.1735, and RMSEA = 0.075 [0.069~0.081]).

As Fig 5 illustrates, mastery experience is the most significant source of EFL learners' speaking self-efficacy and has the greatest explanatory power for enhancing students' confidence in their abilities to deliver fluent oral presentations (LSE), regulate their metacognitive behavior (SRE), maintain emotional control (DSE), and manage their speaking performance (PSE). Notably, physiological and emotional states is the second most significant predictor of EFL learners' speaking self-efficacy in the four EFL-SSES aspects. It has significantly negative contributions to LSE, SRE, DSE, and PSE. This demonstrates that weak physiological and mental states undermine EFL learners' confidence in their speaking ability. Nonetheless, social persuasion is a substantial and positive predictor of DSE and PSE. These results prove that positive feedback from others boosts EFL learners' confidence in delivering speaking performances during English courses. Vicarious experience is a marginally significant predictor of PSE. In other words, modeling has a limited influence on EFL learners' confidence in speaking English.

## 4. General discussion

The purpose of this investigation is to develop and evaluate speaking self-efficacy instruments for EFL learners. These instruments provide a new lens through which to view EFL learners' speaking efficacy and its sources. It is a first attempt to capture the construct of speaking self-efficacy into the development of speaking ability. The items were generated in accordance with the self-efficacy theory and early research on speaking skill. The newly developed EFL-SSES and EFL-SSSES instruments was validated by EFA and CFA, with satisfactory validity and reliability of their factor structure. Notably, the final EFL-SSES is composed of 15 items concerning speaking self-efficacy. Bandura's [6] four sources of self-efficacy are reflected in the final EFL-SSSES by 13 items. EFA and CFA provided significant support for the factorial structure of EFL-SSES and EFL-SSSES. The validation results of these instruments confirm that the four

sources serve as major predictors of the EFL learners' speaking self-efficacy. The path analysis results revealed that four sources of EFL learners' speaking self-efficacy played significant roles in their speaking self-efficacy. In general, the findings not only validated the EFL-SSES and EFL-SSSES psychometrically sound instruments of speaking self-efficacy and its sources, but also demonstrated that their factor structures can be reliably distinguished on both conceptual and empirical grounds. The current study contributes to the social cognitive theory by providing superior evidence from an EFL speaking context in China.

### 4.1 Multiple dimensions of EFL-SSES and EFL-SSSES

The EFL-SSES and EFL-SSSES were created with the primary goal to develop psychometrically sound instruments of speaking self-efficacy. With 15 items on the finalized EFL-SSES, EFA yielded a four-factor structure. This structure was supported by CFA results. LSE refers to an EFL learner's capacity to communicate in a clear, logical, and coherent manner. Given the positive relationship between self-efficacy and English language proficiency, we hypothesize that learners with high scores would be more self-efficacious in their learning process. SRE indicates EFL learners' confidence in their ability to control their metacognitive behavior. According to prior research, students' confidence in self-regulated learning is a significant predictor of academic achievement [13]. We believe that the high-scoring students speak English frequently. DSE reflects students' overall speaking performance in the classroom, including their use of emotional appeal and self-control. As expected, high-scoring students perform better in their speaking assignments. PSE discusses how individuals' perceptions of their abilities affect their behavior in speaking classes. Specifically, the higher-scoring students conduct better in the classroom.

EFA and CFA on the final 13-item EFL-SSSES produced a four-factor solution, in line with the self-efficacy theory [10]. ME refers to cognitive interpretations of prior performance achievements. It has been regarded as the most reliable and substantial source of personal efficacy information [13]. We propose that EFL learners' confidence is boosted when they see their past learning experiences as constructive, while failures reduce their learning confidence. VE demonstrates that when EFL learners lack mastery experience, they obtain necessary knowledge about the performance of respected models in order to evaluate their own potential [44]. It is reasonable to believe that observing role models succeed can favorably reinforce individuals' confidence in performing comparable or identical assignments. SP refers to how EFL learners often depend on important individuals, such as parents, instructors, and peers, for evaluative feedback, judgements, and appraisals about their academic achievement [6]. We believe that when learners acquire efficacy information from others, they are implicitly primed to adjust the effort put on a particular task accordingly. PES indicates that EFL learners would gauge their confidence by recognizing sources of emotional arousal in comparable learning situations. We speculate that EFL learners who are confident in their ability to successfully complete given tasks will experience positive emotions, whereas those who experience negative emotions will experience a decline in self-efficacy, impairing performance and resulting in unsatisfactory learning outcomes [13].

### 4.2 Relationship between the construct and the sources of speaking self-efficacy

The path analysis results demonstrated four sources of speaking self-efficacy are crucial, exerting a strong influence on EFL learners' varied aspects of speaking self-efficacy. The results offer factual support for Bandura's [6] theory of sources of self-efficacy, which have posited

that the four sources are interconnected and exert a substantial influence on the development of self-efficacy.

According to Fig 5, mastery experience is the most influential source of speaking self-efficacy across the four dimensions of speaking self-efficacy, which is consistent with previous findings [13]. EFL learners' interpretations of mastery experiences have an effect on their speaking self-efficacy for comprehending cognitive aspects of the language acquisition process. As predicted, the EFL learners' prior successful speaking experiences influence their self-confidence in presenting oral English, adopting metacognitive strategies, delivering self-controlled speaking presentations, and appraising their speaking competence. Language educators and practitioners can employ instructional techniques, such as task-based language teaching, to promote the development of meaningful experiences and confidence in EFL learners. Then, students' speaking self-efficacy in the domain of mastery experience can be significantly increased.

Notably, the findings indicate that EFL learners' physiological and emotional states are the second most powerful predictor of their linguistic self-efficacy, self-regulatory efficacy, delivery self-efficacy, as well as performance self-efficacy. Since schools give priority to students' reading and writing ability development but neglect listening and speaking [25], students are concerned about the discrepancy between their real speaking performance and what they learned in English textbooks. Many non-native English instructors in China continue to place a premium on the accuracy of vocabulary and grammar rules, as well as repeating dialogues and passages from textbooks [52]. Once students discover incongruence between their prior knowledge and their actual spoken English performance in class, they have elevated feelings of anxiety, tension, and stress related with the classroom speaking activity. For less efficacious students, speaking in the input-poor environment can trigger negative emotional states and a feeling of incompetence. These inadequate emotional arousal is indicative of a lack of ability to accomplish spoken English assignments. In other words, perceived cognitive incongruence influences students' judgment of their own capacities to give speaking performances, then weakening their confidence.

As findings indicate, favorable verbal and nonverbal evaluations from others (social persuasion) contribute to delivery self-efficacy and performance self-efficacy. The encouragement and positive comments received from others can improve EFL learners' engagement in exhibiting advanced cognitive abilities and thoughtfully expressing their viewpoints in a group discussion. In turn, their confidence in oral English is bolstered in these aspects. The path analysis findings imply that social persuasion works in tandem with mastery experience and physiological and emotional states. It accords well with previous research [10, 13] that social persuasion is effective only when combined with other sources affecting self-efficacy.

The efficacy information gained from vicarious experience is not very crucial for EFL learners. It has just a marginal predictive power to self-regulatory efficacy and performance self-efficacy. Due to the prevalence of normative assessment in the classroom, the environment is very competitive. These academic contests and comparisons also act as a benchmark against which learners may assess their self-efficacy. As Chan and Lam [43] previously noted, when students participate in a competitive classroom, their self-efficacy is jeopardized. Consequently, it is unknown how such vicarious experiences from professors or classmates will affect students' capacities and probability of success. The results imply that educators can use constructivist teaching approaches to foster a constructive and collaborative environment in which students collaborate with their classmates and instructors. Additional educational activities emphasizing social interactions and interpersonal connections help students understand the significance of vicarious learning.

Even though some absent relationships in Fig 5 seem to be reasonably present, the predictive effects are diminished while considering the overall model estimated through path analysis

instead of the SEM technique. This implies that the relationships between the two multidimensional constructs are more complex than expected. Future research should be aware of how to thoughtfully formulate models and establish the latent structural relationships based on the findings of this study.

## 5. Implications and limitations

This study successfully confirmed the validity of the speaking self-efficacy scales, while also contributing to the theoretical framework of investigating speaking self-efficacy from a social cognitive perspective. The investigation had educational significance. First, the results indicated that instructors can engage students in realistic English-speaking activities that foster speaking self-efficacy in four aspects. Teachers can design effective teaching methods to boost EFL learners' speaking self-efficacy, especially when considering the sources of self-efficacy. This involves the use of constructivist teaching approaches and task-based language teaching. Additionally, teachers should be aware of students' perceptions of their speaking experiences and work with them to accumulate positive mastery experiences. Moreover, instructors should assist learners to relieve negative emotional arousal, such as frustration, which occur naturally throughout the learning process. Finally, instructors should be attentive to how students interact with one another throughout the speaking activities. These potential interactions give beneficial information for students to foster their speaking self-efficacy.

While the results provided preliminary support for the reliability and validity of the EFL-SSES and EFL-SSSES, some limitations warrant further investigation. First, the sample of this study is primarily composed of advanced English language learners from the same Chinese university, which limits the generalizability of the findings to other cultural or linguistic backgrounds. Replications on the same items of these scales in a different L2 cultural context are still called for. Second, the single research method through self-report data might introduce response biases. Future research is suggested to employ other research approaches, such as emerging methods reviewed by Derakhshan et al. [53], to enrich our understanding of learners' speaking self-efficacy. Moreover, incorporating a variety of possible stakeholders, such as administrators, instructors, and parents, can provide a more systematic view of EFL learners' speaking self-efficacy and its sources. To obtain further understanding, it would be beneficial for researchers to follow the progress of EFL learners' speaking self-efficacy and its sources using a longitudinal study methodology.

## Supporting information

**S1 Appendix. Interview procedure.**
(DOCX)

**S2 Appendix. EFL speaking self-efficacy scale (EFL-SSES).**
(DOCX)

**S3 Appendix. EFL sources of speaking self-efficacy scale (EFL-SSSES).**
(DOCX)

**S1 Data. Raw data.**
(XLSX)

## Author Contributions

**Conceptualization:** Yawen Wang, Peijian Paul Sun.

**Data curation:** Yawen Wang.

**Formal analysis:** Yawen Wang.

**Funding acquisition:** Peijian Paul Sun.

**Investigation:** Yawen Wang.

**Methodology:** Yawen Wang.

**Project administration:** Yawen Wang, Peijian Paul Sun.

**Supervision:** Peijian Paul Sun.

**Writing – original draft:** Yawen Wang.

**Writing – review & editing:** Yawen Wang, Peijian Paul Sun.

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
