## [Decision Letter · Decision Letter 0]

19 Sep 2023

PONE-D-23-22474Development and validation of scales for EFL speaking self-efficacy: Construct, sources, and relationsPLOS ONE

Dear Dr. Sun,

Thank you for submitting your manuscript to PLOS ONE. After careful consideration, we feel that it has merit but does not fully meet PLOS ONE’s publication criteria as it currently stands. Therefore, we invite you to submit a revised version of the manuscript that addresses the points raised during the review process.

The manuscript has been evaluated by three reviewers, and their comments are available below.

The reviewers have raised a number of concerns that need attention. They request additional information on methodological aspects of the study (such as the inclusion of information on the sample size and response rate), revisions to the statistical analyses and they question the internal and external validity of the results reported. Could you please revise the manuscript to carefully address the concerns raised?

We look forward to receiving your revised manuscript.

Kind regards,

Avanti Dey, PhD

Staff Editor

PLOS ONE

“The work was supported by the National Social Science Foundation of China awarded to Dr. Peijian Sun (Grant no. 19CYY008, website http://www.nopss.gov.cn/).”

“The work was supported by the National Social Science Foundation of China (Grant no. 19CYY008).”

“The work was supported by the National Social Science Foundation of China awarded to Dr. Peijian Sun (Grant no. 19CYY008, website http://www.nopss.gov.cn/).”

Reviewers' comments:

Reviewer's Responses to Questions

**Comments to the Author**

1. Is the manuscript technically sound, and do the data support the conclusions?

Reviewer #1: Yes

Reviewer #2: No

Reviewer #3: Yes

2. Has the statistical analysis been performed appropriately and rigorously? 

Reviewer #1: Yes

Reviewer #2: Yes

Reviewer #3: Yes

3. Have the authors made all data underlying the findings in their manuscript fully available?

Reviewer #1: No

Reviewer #2: No

Reviewer #3: Yes

4. Is the manuscript presented in an intelligible fashion and written in standard English?

Reviewer #1: Yes

Reviewer #2: Yes

Reviewer #3: Yes

5. Review Comments to the Author

Reviewer #1: Dear Authors,

I extend my gratitude for crafting a manuscript that introduces two valuable new instruments. Your efforts are commendable. I really enjoyed reading your work. It's evident that your statistical methods are robust, and the discussion is apt. However, prior to advancing towards publication, a few revisions are in order. These suggestions predominantly pertain to the literature review and methodology sections. I encourage you to refer to the detailed feedback I've shared in the attached manuscript file.

I eagerly await the opportunity to engage with your revised manuscript.

With best regards,

Reviewer #2: The manuscript is on the development and validation of an English as a Foreign Language (EFL) Speaking Self-Efficacy Scale (SSES) and its sources which were done in two studies. The authors conducted semi-structured interviews of students to generate items for the scale and used Exploratory Factor Analyses and Confirmatory Factor Analyses to validate the developed scales. They also provide a good review of self-efficacy (SE) and its sources in a general sense. Given that self-efficacy is domain specific, the background of the study needed more information about previous work done in the context of learning a language and what role has SE in language learning. Relevant work is missing from the review. Wang, Kim, Bai, and Hu (2014) validated an English Self-Efficacy Questionnaire in a sample of college students in China. Hong, Hwang, Tai, and Chen (2013) studied English learning anxiety. Wang, Harrison, Cardullo, and Lin (2018) found that English Self-Efficacy and Using-English-to-Learn Self-Efficacy are two distinct constructs. The section on SE measures should focus on speaking SE and other variables studied in relation to it. There may also be other studies that examined the sources of English Learning/Speaking self-efficacy.

The need to measure EFL speaking SE may be presented in relation to learning and practicing the language and how speaking self-efficacy could be measured based on defined dimensions such as grammar, usage, communication, and interaction (see page 2). Are there studies that have looked into these dimensions? The authors operationalized the study constructs well. In the section on the sources of self-efficacy, the authors could also give examples for each source in the context of EFL or language learners, such as in 1st paragraph on page 9. The research questions could be improved (Page 3) for clarity. We know Bandura’s hypothesized sources of SE but what about these sources are the authors interested in?

The scale items are nowhere to be found in the manuscript submitted. The main concern with developing self-efficacy scales is whether the items are true to Bandura’s (2006) guidelines for constructing self-efficacy scales. Wording of the items is key in the measurement of perceived capability. The authors mention that two professors with expertise in SLA reviewed the initial pool of items for content validity. Subjecting the items to review by those with expertise in social cognitive theory to assess whether items reflected self-efficacy was not done. I found it difficult to review this manuscript not knowing the items in the initial pool, what items were deleted, and what items are in the final scale. The authors also did not provide examples of items included in the four factors or in the sources scale. Wang et al. (2014) presented the items in their questionnaire.

The findings in the current study were interesting but without the items included in the scale, the results of the factor analyses are not meaningful. All we know about the scale is the response scale: 1-strongly disagree to 7-strongly agree. The level of agreements does not provide us with what items students are responding to. Even with the results of the EFA and CFA, we are unable to confirm that the items are actually phrased to measure self-efficacy.

The authors used convenience sampling but do not mention participant recruitment. It is unclear whether the questionnaires were administered online or on paper. The reported average time spent to complete the Chinese version was 5 minutes for the 15-item ESSE questionnaire and 5 minutes for the 15-item SSSES questionnaire. One may assume that the students answered an online questionnaire but was that so?

The authors have a sound approach to the study and a solid methodology. Their process for scale development is on point. They examined the internal structure of the items, content and face validity. The authors may perhaps examine concurrent and predictive validity as well. Without the scale items, I am unable to evaluate whether they reflected self-efficacy. The relevance of the study to a larger population of English as a Foreign Language learners is unclear at this point and implications may be premature as both specificity and correspondence in measurement are important in considering self-efficacy’s predictive power. Providing the items in the scales is imperative.

Reviewer #3: Thank you for your submission.

The paper is well-written and argued. However, there are some suggestions that can improve the quality of the paper.

1. Introduction needs to highlight the gaps and significance of the study.

2. The theoretical framework needs to be more comprehensive and more germane studies need to be added.

3. Please elaborate more on the procedure.

4. Data analyses seem robust.

5. Please relate your findings to the theoretical framework.

6. To what extent can this scale be applied in other contexts?

6. PLOS authors have the option to publish the peer review history of their article (what does this mean?). If published, this will include your full peer review and any attached files.

Reviewer #1: **Yes: **Mingzhe Wang

Reviewer #2: No

Reviewer #3: **Yes: **Ali Derakhshan

---

## [Author Response · Author response to Decision Letter 0]

8 Nov 2023

November 1, 2023

PLOS ONE

Dear Editor,

Thank you very much for giving us the opportunity to revise our paper. We would also like to express our gratitude to the reviewers for their constructive feedback, which has greatly contributed to improving the quality of our paper. We have carefully addressed the reviewers’ comments and suggestions, and made the necessary revisions accordingly. Please find below our responses to the reviewers’ comments. We have also submitted our response letter to the system. It would be more reader-friendly to read our responses in our attached word document.

Best regards,

Authors

Review Comments to the Author

Reviewer #1:

Dear Authors,

I extend my gratitude for crafting a manuscript that introduces two valuable new instruments. Your efforts are commendable. I really enjoyed reading your work. It's evident that your statistical methods are robust, and the discussion is apt. However, prior to advancing towards publication, a few revisions are in order. These suggestions predominantly pertain to the literature review and methodology sections. I encourage you to refer to the detailed feedback I've shared in the attached manuscript file.

Our response: Thank you for the recognition of our paper. 

1. The initial section introducing the concept of self-efficacy requires improved clarity in its logical progression. The opening paragraph establishes the notion of self-efficacy and its influence on varying student performances. Subsequently, the second paragraph delves into the domain-specific nature of self-efficacy, focusing on individual levels. The third and concluding paragraph introduces domain specificity in relation to the scarcity of research on listening and speaking self-efficacy. However, the purpose of this section remains unclear. It appears that the authors are discussing the domain specificity characteristic of self-efficacy. But a question arises. Is this the theoretical construct of self-efficacy that the authors attempt to introduce or explain? If the intention is to introduce self-efficacy within the framework of social-cognitive theory, there is a notable absence of an elaboration on the connection between the two. Crucially, self-efficacy plays a central role in Bandura's social-cognitive theory. Consequently, a more comprehensive exposition of this theory and the rationale behind self-efficacy's pivotal position would be apt.

To enhance the argumentation, two recommendations are proposed. First, a more robust logical progression must be established to reinforce the argument. Second, a more comprehensive exploration of the social-cognitive theory and its relationship with self-efficacy is necessary. This approach will effectively offer readers with a solid theoretical foundation.

Our response: Thank you for this valuable comment. In terms of logical progression, we have modified and refined the introduction and the literature review to enhance its logical flow and coherence. Please find the highlighted revisions in the two sections. In terms of connection between the social-cognitive theory and self-efficacy, we have revised accordingly to enhance their connections on pages 3 and 4. That is, “According to social cognitive theory, learners are agents who “draw on their knowledge and cognitive and behavioral skills to monitor their actions [and] enlist cognitive guides and self-incentives to produce desired results” [1]…… Bandura [12] argues that the operational function of self-efficacy depends on an essential belief linking human agency and efficacy beliefs.” 

2. May be better to provide more explanation on this. Why moderate levels of self-efficacy beliefs is beneficial to language learning?

Our response: Thank you for the feedback. Together with other reviewers’ comments, we have substantially revised our literature review. The statement in terms of EFL learners’ moderate levels of self-efficacy has been deleted.

3. I acknowledge that the intention behind this sentence is to establish the rationale for the current research. However, its placement appears abrupt in the flow of the narrative. A noticeable shift occurs at this juncture, transitioning from a short introduction to self-efficacy in the realm of EFL research to advocating for the development of a speaking self-efficacy scale and introducing existing ones. From my perspective, the inclusion of speaking self-efficacy measurement doesn't seem seamlessly integrated into the overall presentation of the "construct of speaking self-efficacy."

It is my viewpoint that these arguments might find a more fitting placement in the subsequent section dedicated to "self-efficacy measures." Consequently, I propose two revisions for this section. Firstly, a more comprehensive exploration of relevant knowledge and literature concerning the construct of speaking self-efficacy should be included. Secondly, the consideration of relocating the discussion on speaking self-efficacy measurement to the ensuing section could enhance the overall coherence of the presentation.

Our response: Thank you for this valuable comment. In terms of the construct of speaking self-efficacy, we have presented a thorough review in Section 2.2 on pages 5-7 to the best of our knowledge. One of the aims of our study is to establish an EFL speaking self-efficacy construct to enrich the limited literature on the construct of speaking self-efficacy.

In terms of the coherence of our paper presentation, we have made substantial revision to enhance the logic and flow. Please find the highlighted part in Section 2.2 for our revision. 

4. More elaborations on "the mix use of general and task-specific self-efficacy in EFL research" are needed if the authors attempt to arrive to the conclusion of "the findings of self-efficacy display inconsistency". The above presentation of various self-efficacy scales, while informative, does not adequately contribute to the authors' objective of arriving at this inconsistency in self-efficacy findings.

Our response: Thank you for this valuable comment. We have reorganized and rewritten the last paragraph of Section 2.2 to enhance the arguments. Please find the highlighted paragraph for our revision (pp. 7-8).

5. Please include the interview data and the explanations of how the data contribute to generating the initial items into the Appendix and upload them to the system. Take utmost care in safeguarding the participants' anonymity while uploading the appendix materials.

Our response: We thank the reviewer for this valuable comment and would like to draw his/her attention to the added supplementary information in Appendix A. There, the outline of semi-structured interview is provided. For that, we added in Section 3.1 (p. 11): “The interview protocol was designed based on previous literature [9,39]. S1 Appendix presents the analysis procedure of the interview data, which helped to generate the initial items.” Due to length constraints, we will not be delving into the interviews’ analysis in detail here.

6. What are the hypothesized factors? Please introduce them in the previous section where the authors explained the process of intial items generation.

Our response: We thank the reviewer for this valuable comment. The hypothesized factors are the four sources of self-efficacy based on Bandura’s social cognitive theory. We illustrated the item generation process in Section 3.2.1 (p. 18). The process of item generation was informed by Bandura’s self-efficacy theory. According to the theory, there are four sources of self-efficacy, including mastery experience, vicarious experience, social persuasion, physiological and emotional states. By drawing on the extant literature on sources of self-efficacy, an initial pool of 30 items was constructed. 

7. Please offer a brief explanation of using a promax rotaion method, as it is very important for an EFA.

Our response: We thank the reviewer for this valuable comment. We chose the promax rotaion method because of the following reasons. According to Loewen & Gonulal, (2015, p. 197), the most appropriate type of rotation in SLA research is generally oblique rotation. Since four factors arguably tapped into different aspects of speaking self-efficacy, oblique rotation is considered to be more appropriate. We decided to use promax rotation in the EFAs conducted in the research. 

8. Please also include the decision process of how the initial items were generated into the Appendix.

Our response: We thank the reviewer for this valuable comment. We have added the process of item revision and the reasons behind it in S1 Appendix: “During the item generation phase, semi-structured interview was administered to a total of 10 students. participants were asked to respond in their first language (Chinese). To analyze the transcribed interviews, the data were analyzed by the first author by omitting data which were not relevant to speaking self-efficacy. They then compared their lists of items and together created a 30-item questionnaire.”

I eagerly await the opportunity to engage with your revised manuscript.

Our response: Thank you for the encouragement.

Reviewer #2: 

The manuscript is on the development and validation of an English as a Foreign Language (EFL) Speaking Self-Efficacy Scale (SSES) and its sources which were done in two studies. The authors conducted semi-structured interviews of students to generate items for the scale and used Exploratory Factor Analyses and Confirmatory Factor Analyses to validate the developed scales. They also provide a good review of self-efficacy (SE) and its sources in a general sense. Given that self-efficacy is domain specific, the background of the study needed more information about previous work done in the context of learning a language and what role has SE in language learning. Relevant work is missing from the review. Wang, Kim, Bai, and Hu (2014) validated an English Self-Efficacy Questionnaire in a sample of college students in China. Hong, Hwang, Tai, and Chen (2013) studied English learning anxiety. Wang, Harrison, Cardullo, and Lin (2018) found that English Self-Efficacy and Using-English-to-Learn Self-Efficacy are two distinct constructs. The section on SE measures should focus on speaking SE and other variables studied in relation to it. There may also be other studies that examined the sources of English Learning/Speaking self-efficacy.

Our response: We thank the reviewer for this valuable comment. We have added these relevant works in the literature review. 

The need to measure EFL speaking SE may be presented in relation to learning and practicing the language and how speaking self-efficacy could be measured based on defined dimensions such as grammar, usage, communication, and interaction (see page 2). Are there studies that have looked into these dimensions? The authors operationalized the study constructs well. In the section on the sources of self-efficacy, the authors could also give examples for each source in the context of EFL or language learners, such as in 1st paragraph on page 9. The research questions could be improved (Page 3) for clarity. We know Bandura’s hypothesized sources of SE but what about these sources are the authors interested in?

Our response: Thank you for your thought-provoking comments. In terms of studies that have looked into these dimensions, we have reviewed, to the best of our knowledge, all direct relevant studies on speaking self-efficacy. Please find the revised Section 2.2 for details. 

 With regard to examples for each source, we have modified the illustrations of each source in the context of EFL. Please find the highlighted sentences in Section 2.3 for details.

 In terms of research questions, we have partly revised them for the clarity sake. More importantly, we have refined the introduction section to resonate it with the proposed research questions to enhance the logic and flow between the identified research gaps and the research questions. Please find the highlighted in the introduction for details.

 In terms of speaking self-efficacy sources, we only drew on Bandura’s hypothesized sources of SE because studies with theoretical foundations will be more robust. We believe there may be other sources. However, this is not the focus of our current study. Future research may consider delving into this matter. In brief, identifying the relevant sources of speaking self-efficacy that students may encounter in English speaking classrooms may better enhance their speaking self-efficacy.

The scale items are nowhere to be found in the manuscript submitted. The main concern with developing self-efficacy scales is whether the items are true to Bandura’s (2006) guidelines for constructing self-efficacy scales. Wording of the items is key in the measurement of perceived capability. The authors mention that two professors with expertise in SLA reviewed the initial pool of items for content validity. Subjecting the items to review by those with expertise in social cognitive theory to assess whether items reflected self-efficacy was not done. I found it difficult to review this manuscript not knowing the items in the initial pool, what items were deleted, and what items are in the final scale. The authors also did not provide examples of items included in the four factors or in the sources scale. Wang et al. (2014) presented the items in their questionnaire.

Our response: We thank the reviewer for this valuable comment. Due to the word limit, we were not able to present items in detail in our manuscript. However, we have added the original items in the S2 Appendix and S3 Appendix, as well as the two Chinese-version fully rating scales with labels. 

The findings in the current study were interesting but without the items included in the scale, the results of the factor analyses are not meaningful. All we know about the scale is the response scale: 1-strongly disagree to 7-strongly agree. The level of agreements does not provide us with what items students are responding to. Even with the results of the EFA and CFA, we are unable to confirm that the items are actually phrased to measure self-efficacy.

Our response: We thank the reviewer for this valuable comment. As we have responded earlier on, the item details can be found in the S2 Appendix and S3 Appendix. 

The authors used convenience sampling but do not mention participant recruitment. It is unclear whether the questionnaires were administered online or on paper. The reported average time spent to complete the Chinese version was 5 minutes for the 15-item ESSE questionnaire and 5 minutes for the 15-item SSSES questionnaire. One may assume that the students answered an online questionnaire but was that so?

Our response: Thank you for the feedback. We have added the sampling method in the Participants parts of Section 3.1.2, Section 3.1.3, Section 3.2.2, and Section 3.2.3 for clarification. That is, “Questionnaires were administered during their college English course.”

The authors have a sound approach to the study and a solid methodology. Their process for scale development is on point. They examined the internal structure of the items, content and face validity. The authors may perhaps examine concurrent and predictive validity as well. Without the scale items, I am unable to evaluate whether they reflected self-efficacy. The relevance of the study to a larger population of English as a Foreign Language learners is unclear at this point and implications may be premature as both specificity and correspondence in measurement are important in considering self-efficacy’s predictive power. Providing the items in the scales is imperative.

Our response: Thank you for your thought-provoking comment. Please find the item details in the S2 Appendix and S3 Appendix. In terms of the concurrent and predictive validity, we understand it will make the scale development more robust. However, due to the paper length constraint, we are not able to present details on the matter of concurrent and predictive validity. We will provide these details in our future work. 

Reviewer #3:

Thank you for your submission.

The paper is well-written and argued. However, there are some suggestions that can improve the quality of the paper.

Our response: Thank you for the recognition of our paper. 

1. Introduction needs to highlight the gaps and significance of the study.

Our response: Thank you this valuable comment. We have reorganized and rewritten the introduction to enhance the significance of the study. Please find the highlighted introduction section for our revision.

2. The theoretical framework needs to be more comprehensive and more germane studies need to be added.

Our response: We thank the reviewer for this valuable comment. We have added more relevant studies in the literature review. Please find the highlighted on Section 2 (pp. 4-9) for details.

3. Please elaborate more on the procedure.

Our response: We thank the reviewer for this comment. We have modified the procedure according to your suggestion. Please find the highlighted on Section 3.1.1 and 3.1.2 (pp. 11-12) for details.

4. Data analyses seem robust.

Our response: Thank you for the recognition of our data analyses. 

5. Please relate your findings to the theoretical framework.

Our response: Thank you this valuable comment. We have added the contribution of our results to the theoretical framework. Please find the highlighted part in Section 4 (p. 25) and Section 4.2 (p. 27) for our revision.

6. To what extent can this scale be applied in other contexts?

Our response: We thank the reviewer for this valuable comment. The great potential of this study lies in the application of speaking self-efficacy beliefs in the language teaching/learning classroom. The survey can help teachers of English in China to make diagnostic assessments of their students’ speaking self-efficacy. Teachers are recommended to identify specific self-efficacy conditions of their students. Having acknowledged students’ confidence in their speaking capabilities, teachers are also encouraged to design instructional activities to nurture high levels of speaking self-efficacy beliefs. Please find the highlighted on Section 5 (p. 30) for details.

---

## [Decision Letter · Decision Letter 1]

27 Nov 2023

PONE-D-23-22474R1Development and validation of scales for EFL speaking self-efficacy: Constructs, sources, and relationsPLOS ONE

Dear Dr. Sun,

Thank you for submitting your manuscript to PLOS ONE. After careful consideration, we feel that it has merit but does not fully meet PLOS ONE’s publication criteria as it currently stands. Therefore, we invite you to submit a revised version of the manuscript that addresses the points raised during the review process. Please thoroughly address the concerns raised by the reviewers, making any necessary revisions to your manuscript. Additionally, I have noted a discrepancy in the author order between your manuscript and our system records. To ensure consistency, kindly verify and, if needed, correct this in the system or in the manuscript. In case the system's authorship record requires modification, please avert potential authorship disputes by providing a signed statement from all authors, agreeing to the changes. This statement should be scanned and uploaded to the system for our records.

We look forward to receiving your revised manuscript.

Kind regards,

Mingzhe Wang, Ph.D.

Guest Editor

PLOS ONE

Journal Requirements:

Reviewers' comments:

Reviewer's Responses to Questions

**Comments to the Author**

1. If the authors have adequately addressed your comments raised in a previous round of review and you feel that this manuscript is now acceptable for publication, you may indicate that here to bypass the “Comments to the Author” section, enter your conflict of interest statement in the “Confidential to Editor” section, and submit your "Accept" recommendation.

Reviewer #3: All comments have been addressed

Reviewer #4: All comments have been addressed

2. Is the manuscript technically sound, and do the data support the conclusions?

Reviewer #3: Yes

Reviewer #4: Yes

3. Has the statistical analysis been performed appropriately and rigorously? 

Reviewer #3: Yes

Reviewer #4: Yes

4. Have the authors made all data underlying the findings in their manuscript fully available?

Reviewer #3: Yes

Reviewer #4: Yes

5. Is the manuscript presented in an intelligible fashion and written in standard English?

Reviewer #3: Yes

Reviewer #4: Yes

6. Review Comments to the Author

Reviewer #3: Thank you for the revised file. I am happy with the revised file. The authors can proofread their paper.

Reviewer #4: Dear Authors,

I have thoroughly reviewed your manuscript titled "Development and validation of scales for EFL speaking self-efficacy: Constructs sources and relations." Your study offers valuable insights into the area of EFL speaking self-efficacy and contributes significantly to the existing body of knowledge. However, there are several areas where the manuscript could be strengthened.

Consistency in Terminology: Maintain consistent use of key terms throughout the paper. For example, terms like "self-efficacy," "speaking self-efficacy," and "EFL speaking self-efficacy" should be used consistently to avoid confusion. It is recommended that you refer to this article when it comes to the concept of self-efficacy.

Han, Y., & Wang, Y. (2021). Investigating the correlation among Chinese EFL teachers’ self-efficacy, reflection, and work engagement. Frontiers in Psychology 12. 763234. https://doi.org/10.3389/fpsyg.2021.763234.

Comprehensiveness: Ensure that the literature review covers all relevant studies, especially recent research, to demonstrate the current state of knowledge in the field. Instead of only summarizing previous studies, critically evaluate them to highlight their contributions and limitations in relation to your research.

Methodology Section: Provide more detail on the sampling strategy to clarify how representativeness was ensured or the limitations. In the item generation and validation process, consider elaborating on how you ensured the cultural relevance of the items, especially since the study focuses on Chinese EFL learners.

Limitations section: Acknowledge the limitations of your study more explicitly, such as the reliance on self-report measures, which might introduce response biases. This study focuses on a specific population (Chinese EFL learners) and context, which may limit the generalizability of the findings to other cultural or linguistic backgrounds. You can supplement it in Limitations. It is recommended that you could refer to this article

Derakhshan, A., Wang, Y. L., Wang, Y. X., & Ortega-Martín, J. L. (2023). Towards Innovative Research Approaches to Investigating the Role of Emotional Variables in Promoting Language Teachers’ and Learners’ Mental Health. International Journal of Mental Health Promotion, 25 (7): 823-832. doi:10.32604/ijmhp.2023.029877.

Language and Style: Perform a thorough proofreading to catch any minor grammatical or syntactical errors. Ensure consistency in terminology and style throughout the paper.

Overall, your research is promising and with these suggested improvements, it has the potential to make a more impactful contribution to the field of EFL education.

Sincerely,

7. PLOS authors have the option to publish the peer review history of their article (what does this mean?). If published, this will include your full peer review and any attached files.

Reviewer #3: No

Reviewer #4: **Yes: **Yongliang Wang

---

## [Author Response · Author response to Decision Letter 1]

2 Jan 2024

Please find the uploaded file entitled "response to comments" for our detailed response to reviewers' comments.

---

## [Editor Report · Decision Letter 2]

8 Jan 2024

Development and validation of scales for speaking self-efficacy: Constructs, sources, and relations

PONE-D-23-22474R2

Dear Dr. Sun,

We’re pleased to inform you that your manuscript has been judged scientifically suitable for publication and will be formally accepted for publication once it meets all outstanding technical requirements.

Kind regards,

Mingzhe Wang, Ph.D.

Guest Editor

PLOS ONE
---

## [Editor Report · Acceptance letter]

14 Jan 2024

PONE-D-23-22474R2 

PLOS ONE

Dear Dr. Sun, 

I'm pleased to inform you that your manuscript has been deemed suitable for publication in PLOS ONE. Congratulations! Your manuscript is now being handed over to our production team.

Kind regards, 

on behalf of

Dr. Mingzhe Wang 

Guest Editor

PLOS ONE